# The Efficacy of S-1 as Adjuvant Chemotherapy for Resected Biliary Tract Carcinoma: A Propensity Score-Matching Analysis

**DOI:** 10.3390/jcm10050925

**Published:** 2021-03-01

**Authors:** Yoichi Miyata, Ryota Kogure, Akiko Nakazawa, Rihito Nagata, Tetsuya Mitsui, Riki Ninomiya, Masahiko Komagome, Akira Maki, Nobuaki Kawarabayashi, Yoshifumi Beck

**Affiliations:** 1Department of Surgery, Asahi General Hospital, 1326 I, Asahi-shi, Chiba 289-2511, Japan; 2Department of Surgery, National Defence Medical College, 3-2 Namiki, Tokorozawa-shi, Saitama 359-8513, Japan; nakazawaa-sur@h.u-tokyo.ac.jp; 3Department of Hepatobiliary Pancreatic Surgery, Saitama Medical Centre, Kamoda 1981, Kawagoe-shi, Saitama 350-5500, Japan; ballershigh23and1@yahoo.co.jp (R.K.); tmitchiex@gmail.com (T.M.); rickynino9222@gmail.com (R.N.); komagome@saitama-med.ac.jp (M.K.); akiramaki.md@gmail.com (A.M.); ybeck@saitama-med.ac.jp (Y.B.); 4Hepato-Biliary-Pancreatic Surgery Division, Department of Surgery, Graduate School of Medicine, The University of Tokyo, Bunkyo-ku, Tokyo 13-8655, Japan; NAGATAR-SUR@h.u-tokyo.ac.jp; 5Department of Surgery, Gyoda General Hospital, Mochida 376, Gyoda-shi, Saitama 361-0056, Japan; kawarabayashi@gyoda-hp.or.jp

**Keywords:** adjuvant chemotherapy, biliary tract carcinoma, propensity score matching, retrospective, S-1

## Abstract

Even though S-1 is a widely used chemotherapeutic agent, there is no evidence for its use in an adjuvant setting for biliary tract carcinoma (BTC). Patients who underwent surgical treatment for BTC between August 2007 and December 2018 were selected. Propensity score matching was performed between patients who received S-1 as adjuvant chemotherapy (S-1 group) and those who underwent surgical treatment alone (observation group). Of 170 eligible patients, 38 patients were selected in each group after propensity score matching. Among those in the matched cohort, both the median recurrence-free survival (RFS) and overall survival (OS) in the S-1 group were significantly longer than those in the observation group (RFS, 61.2 vs. 13.1 months, *p* = 0.033; OS, not available vs. 28.2 months, *p* = 0.003). A multivariate analysis of the OS revealed that perineural invasion and adjuvant S-1 chemotherapy were independent prognostic factors. According to a subgroup analysis of the OS, the S-1 group showed significantly better prognoses than the observation group among patients with perineural invasion (*p* < 0.001). S-1 adjuvant chemotherapy might improve the prognosis of BTC, especially in patients with perineural invasion.

## 1. Introduction

Biliary tract carcinoma (BTC) is a relatively rare cancer worldwide [1]. According to the World Health Organization classification, BTC includes perihilar and distal extrahepatic bile duct carcinoma and gallbladder carcinoma [2]. The surgical procedure for BTC depends on the location of the lesion. For example, major hepatectomy with extra bile duct resection is performed for perihilar carcinoma, and pancreaticoduodenectomy is the most common approach for distal cholangiocarcinoma. Even though radical resection is required to completely remove the tumour, the recurrence rate is reported to be high, around 50% [3], and the overall survival (OS) rate remains poor.

The benefit of adjuvant chemotherapy after the surgical treatment of several advanced cancers, such as gastric cancer [4,5,6], colon cancer [7,8], and pancreatic cancer [9], is well established. Various adjuvant chemotherapy regimens are reported to improve the prognosis of patients with such cancers. However, in the case of BTC, the few large randomized trials on adjuvant chemotherapy conducted to date have produced unpromising results [10,11] and the efficacy of adjuvant chemotherapy for BTC remains unknown.

S-1 is an oral anti-cancer drug consisting of tegafur, 5-chloro-2,4-dihydroxypridine, and potassium oxonate [12,13]. The benefit of S-1 as an adjuvant chemotherapy has been reported for gastric cancer [4] and pancreatic cancer [9]. For BTC, several studies have shown that the efficacy of S-1 adjuvant chemotherapy varies [14,15], and its effectiveness remains debatable.

The aim of the present study was to retrospectively investigate the efficacy of S-1 administration as adjuvant chemotherapy after the surgical treatment of BTC. Because the patients with advanced carcinoma received adjuvant chemotherapy, we performed propensity score matching to reduce the inherent bias.

## 2. Materials and Methods

### 2.1. Patient Selection

Charts from two institutions were reviewed to select the patients who had undergone surgical treatment for BTC at Saitama Medical Centre and Gyoda General Hospital between August 2007 and December 2018. All the selected patients were pathologically diagnosed with BTC, including gallbladder carcinoma, perihilar cholangiocarcinoma, and distal cholangiocarcinoma. Patients who had received chemotherapies other than S-1 before and/or after their surgical treatment, had undergone R2 resection, or had not been able to receive S-1 adjuvant chemotherapy because they had died from postoperative complications within 90 days after surgery were excluded.

All the clinical, laboratory, radiologic, and pathological data were collected from electronic medical records. The study was conducted according to the guidelines of the Declaration of Helsinki and approved by the Institutional Review Board of Saitama Medical Centre, Saitama Medical University (No. 2002), and Gyoda General Hospital (No. 2019-1).

### 2.2. Treatment Strategy and Follow Up

The surgical treatment strategy was planned in accordance with the clinical status, such as pancreaticoduodenectomy for distal cholangiocarcinoma or major hepatectomy with extra bile duct resection for perihilar cholangiocarcinoma and gallbladder carcinoma. All the patients underwent adequate regional lymph node dissection, including the removal of hilar and pericholedochal nodes in the hepatoduodenal ligament, posterior and anterior pancreaticoduodenal nodes, and nodes along the common hepatic artery [16]. Intraoperative pathological examination of the proximal and/or distal biliary tract margins was performed to confirm carcinoma-free margins using frozen tissue sections. If the biliary tract margin was positive for carcinoma, then additional biliary tract resection was performed until the margin was free (or to the maximum extent possible) from carcinoma.

Patients in each institution were followed up after their surgical treatment every 3 to 6 months, which consisted of basic blood examinations, including the carbohydrate antigen 19-9 (CA19-9) level, and imaging examinations were usually performed with contrast-enhanced computed tomography. Additional imaging examinations were performed if recurrence was suspected. The end of the follow-up period was set as March 2019 or the date of death.

### 2.3. Administration Criteria of Adjuvant S-1 Chemotherapy

We considered the administration of S-1 (TS-1; Taiho, Tokyo, Japan) as adjuvant chemotherapy for patients to whom any of the following pathological findings applied: positive for lymphatic invasion and/or venous invasion and/or perineural invasion; positive for lymph node metastases, microscopic residual tumor, T status of T3 or T4. The patients also satisfied all of following criteria: Eastern Cooperative Oncology Group performance status (PS) of less than 2, adequate bone marrow function (leukocyte count ≥ 3000 cells per cubic millimeter, hemoglobin concentration ≥ 8.0 g/dL, and platelet count ≥ 100,000 cells per cubic millimeter), adequate liver function (total bilirubin concentration ≤ 2.0 mg/dL, asparate aminotransferase concentration ≤ 100 IU/L, and alanine aminotransferase concentration ≤ 100 IU/L), adequate renal function (serum creatinine concentration ≤ 1.5 mg/dL).

These patients received oral S-1 twice daily at a dose matched to their body surface area (BSA) as follows: BSA < 1.25 m^2^, 80 mg/day; 1.25 m^2^ ≤ BSA < 1.50 m^2^, 100 mg/day; and 1.50 m^2^ ≤ BSA, 120 mg/day [13]. S-1 was administered for 28 days, followed by 14 days of rest in each 42-day cycle. Adjuvant S-1 chemotherapy was performed as long as possible unless the patients’ condition were intolerable such as PS was higher than 2, or liver and/or renal disfunction. The patients who experienced recurrence were given adequate treatment, including the best supportive care.

### 2.4. Statistical Analysis

All the pathological diagnoses were recorded in accordance with the 8th edition of the Union for International Cancer Control TMN classification [16]. OS was defined as the interval from the date of surgical treatment to the date of death from any cause or the end of the follow-up period. Recurrence-free survival (RFS) was defined as the interval from the date of surgical treatment to the date of confirmed recurrence. The interval from the surgical treatment to the date of death or end of the follow-up period for patients without recurrence was also defined as RFS.

Continuous data were expressed as the median with range. Quantitative and categorized variables were compared using Wilcoxon’s rank-sum test and the chi-squared test, respectively. RFS and OS were estimated using the Kaplan–Meier method, and differences in survival curves were compared using the log-rank test. The multivariate analysis of OS was performed using a Cox proportional-hazards model to the factors statistically significant on univariate analysis, and the results were expressed as the hazard ratio (HR) and 95% confidence interval (95% CI).

Potential co-variables included in the propensity score matching were age, CA19-9 level, tumour (perihilar and distal extrahepatic bile duct carcinoma and gallbladder carcinoma), tumour differentiation, lymphatic invasion, venous invasion, perineural invasion, T status, N status, R status, and postoperative complications in accordance with the Clavien–Dindo classification [17]. Propensity scores were estimated using a logistic regression model, and the C-statistic for evaluating the goodness of fit was calculated. A one-to-one nearest-neighbour matching algorithm was applied with a calliper of 0.2.

*p* values ≤ 0.050 were considered statistically significant. All the statistical analyses were performed using JMP software (version 9.0.0; SAS Institute, Cary, NC, USA).

## 3. Results

### 3.1. Patient Characteristics in the Entire Cohort

During the study periods, 252 patients underwent surgical treatment for BTC. One patient who received chemotherapy before surgery, 23 who received adjuvant chemotherapies other than S-1, 24 treated with R2 resection, and 19 who died within 90 days after surgery without receiving S-1 adjuvant chemotherapy were excluded. Three patients who did not meet the administration criteria received adjuvant S-1 chemotherapy and 12 patients who met the administration criteria received surgical treatment alone were also excluded. Finally, 170 patients were designated as the entire cohort (Figure 1).

The median age of the entire cohort was 74 (range, 42 to 90) years, and 106 (62%) were male. There were 116 (68%) cases of cholangiocarcinoma (49 (29%) hilar cholangiocarcinoma and 67 [39%] distal cholangiocarcinoma) and 54 (32%) cases of gallbladder carcinoma. Hepatectomy was required in 51 (30%) patients (35 (21%) major hepatectomy and 16 (9%) hepatopancreaticoduodenectomy) and 70 patients underwent pancreaticoduodenectomy. Lymphatic invasion, venous invasion, perineural invasion, and lymph node metastases were observed in 87 (51%), 102 (60%), 113 (66%), and 71 (42%) patients, respectively. R0 resection was achieved in 122 (72%) patients. S-1 adjuvant chemotherapy was administered in 77 (45%) patients, and the median duration of S-1 administration was 10.6 (range, 1.9 to 59.3) months. The profiles and tumour characteristics of the patients who received S-1 adjuvant chemotherapy (S-1 group) and surgical treatment alone (observation group) are shown in Table 1. The median age of the S-1 group was significantly lower than that of the observation group (*p* < 0.001). Lymphatic invasion, venous invasion, and perineural invasion were observed significantly more often in the S-1 group than in the observation group (*p* = 0.001, 0.001, and 0.005, respectively). The proportion of patients with a T status of “T3 and T4” and N1 disease was also higher in the S-1 group (*p* < 0.001 for each). The R0 resection rate was comparable between two groups (*p* = 0.255). 

The median RFS and OS of the entire cohort were 34.0 and 86.7 months, respectively. The median length of the follow-up interval was 50.6 months. Kaplan–Meier curves of the RFS and OS are shown in Appendix A.

### 3.2. Patient Characteristics and Survival in the Matched Cohort

After propensity score matching, 76 patients (38 in both the S-1 and observation groups) were selected. The C-statistic for the goodness of fit was 0.818. Table 2 shows the profiles and tumour characteristics of the patients in each group in the matched cohort. Hilar cholangiocarcinoma, distal cholangiocarcinoma, and gallbladder carcinoma were present in 11 (29%), 18 (47%), and 9 (24%) patients in the S-1 group, respectively, and in 10 (26%), 17 (45%), and 11 (29%) patients in the observation group, respectively (*p* = 0.871). 

The median interval from surgical treatment to the initiation of adjuvant S-1 chemotherapy was 63 (range, 21 to 146) days, and the median duration of S-1 administration was 11.1 (range, 1.9 to 59.3) months.

The both median RFS and OS was significantly longer in the S-1 group than observation group (RFS, 61.2 vs. 13.1 months, *p* = 0.033; OS, not available vs. 28.2 months, *p* = 0.003) (Figure 2).

### 3.3. Univariate and Multivariate Analysis of OS in the Matched Cohort

The univariate and multivariate analysis of OS in the matched cohort is shown in Table 3. According to the univariate analysis, adjuvant S-1 chemotherapy as well as venous invasion and perineural invasion were significant predictors. The multivariate analysis revealed the presence of perineural invasion (Hazard ratio [HR] = 6.038, 95% CI, 1.709–29.153, *p* = 0.004) without adjuvant S-1 chemotherapy (HR = 4.370, 95% CI, 1.989–10.298, *p* < 0.001) was an independent poor prognostic factor. 

### 3.4. Subgroup Analysis of the Prognostic Impact of S-1 Adjuvant Chemotherapy 

To evaluate the prognostic impact of adjuvant S-1 chemotherapy for patients with poor prognostic factors of perineural invasion, we compared the OS of the patients with perineural invasion between the S-1 group and the observation group. The profiles and tumour characteristics of the patients with perineural invasion are shown in Appendix A. The median OS of the patients with perineural invasion in S-1 group was significantly better than that of the observation group (not available vs. 18.1 months, *p* < 0.001) (Figure 3).

## 4. Discussion

This study investigated the postoperative outcomes of BTC resection with the administration of S-1 as adjuvant chemotherapy. Given that patients considered to be at a high risk of recurrence would likely receive adjuvant chemotherapy, we performed a propensity score-matching analysis to reduce patient selection bias. In our matching cohort, both the RFS and OS of the patients in the S-1 group were significantly longer than those in the observation group. Furthermore, adjuvant S-1 chemotherapy might contribute to the improved prognosis of patients with perineural invasion.

Several studies have reported on adjuvant chemotherapy for BTC [10,11,15,18,19,20,21,22,23,24,25,26]. The multicentre randomized phase III trial PRODIGE 12-ACCORD 18 conducted by a French group failed to show the efficacy of gemcitabine and oxaliplatin (GEMOX) in treating BTC patients in an adjuvant setting [10]. Additionally, the randomized phase III trial BCAT from Japan also failed to show a significant efficacy of adjuvant gemcitabine chemotherapy [11]. The results of the present study differ from those reported by these two large randomized studies. One explanation may be related to the different adjuvant chemotherapeutic agents used, whereby S-1 appears to achieve a better outcome when compared with gemcitabine in an adjuvant setting [14].

The efficacy of S-1 as adjuvant chemotherapy is well established for gastric cancer [4] and pancreatic cancer [9]. Regarding BTC, some studies showed that adjuvant S-1 improved the prognosis [14,15]. Given that S-1 contains the 5-fluorouracil (5-FU) prodrug tegafur [13], previous studies using 5-FU as adjuvant chemotherapy for BTC [21,23] have also suggested the potential efficacy of S-1. Recently, a multicentre randomized phase III trial of adjuvant chemotherapy for BTC (BILCAP) reported the efficacy of capecitabine, one of the prodrugs of 5-FU, with an OS of 53 months in the adjuvant group versus 36 months in the observation group (*p* = 0.028) [24]. It was reported that the allelic variants of CYP2A6, which is the metabolic enzyme of 5FU, were different between Caucasian and East Asian populations, but the pharmacokinetics of S-1 were not significantly different [27]. All of these previous studies support our current positive data for the use of S-1 in an adjuvant setting.

There are various reports on the prognostic factors after resection for BTC [28,29,30]. Perineural invasion [31,32] was reported to be one of the poor prognostic factors. In our series, the patients with perineural invasion showed a poor prognosis and thus might benefit from adjuvant chemotherapy with S-1. Further studies are required to investigate the extent of benefit from adjuvant chemotherapy for BTC.

The present study has several limitations. The first was its retrospective nature. Although we analysed our data using propensity score matching, some selection bias may have remained. Second, our series contained a heterogeneous group of BTC patients and a small sample size. A future study with a homogeneous group of BTC patients and a larger sample size is required to confirm our results. Finally, the administration protocol of adjuvant S-1 was not unified, particularly the duration of administration. Further controlled prospective research is necessary, and the final results of the JCOG 1202 study [33], a randomized phase III trial of adjuvant S-1 therapy versus observation alone in resected BTC patients, are awaited.

In conclusion, we reported the efficacy of S-1 as adjuvant chemotherapy after the resection of BTC using a propensity score matching analysis, and our results suggest that this approach might improve patients’ prognoses, especially in patients with perineural invasion.

## Figures and Tables

**Figure 1 jcm-10-00925-f001:**
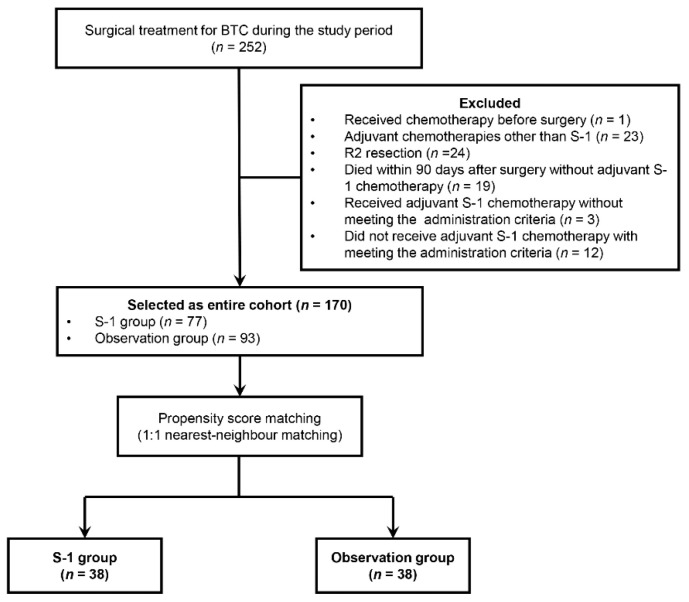
Flow chart of the patients included the study.

**Figure 2 jcm-10-00925-f002:**
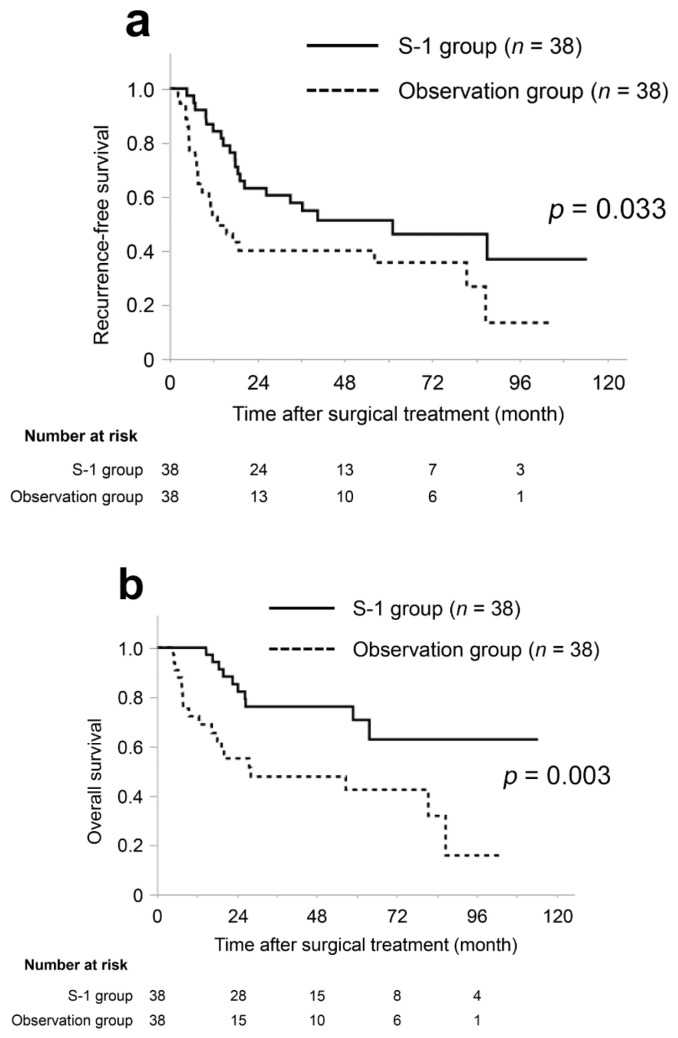
Kaplan–Meier curves of recurrence-free survival (**a**) and overall survival (**b**) in the matched cohort. The survival duration in the S-1 group was significantly longer than that in the observation group both in recurrence-free survival and overall survival (*p* = 0.033 and *p* = 0.003, respectively).

**Figure 3 jcm-10-00925-f003:**
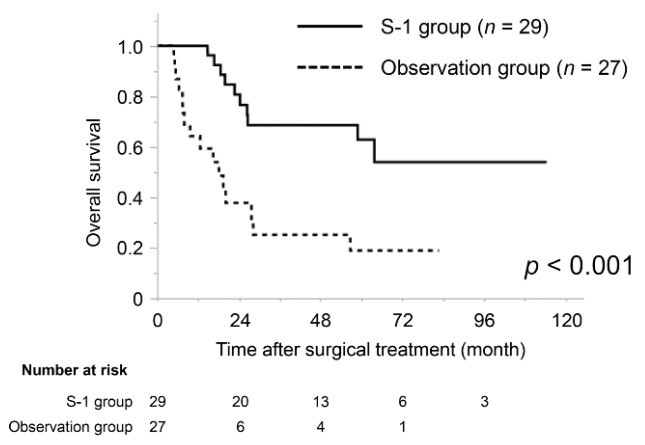
Kaplan–Meier curves for the overall survival of matched cohort patients with perineural invasion. The overall survival of the S-1 group was significantly better than that of the observation group (*p* < 0.001).

**Table 1 jcm-10-00925-t001:** Profiles and tumour characteristics of the patients in each group of the entire cohort.

	S-1 Group *n* = 77	Observation Group *n* = 93	*p* Value
Age [y]	70 (44–87)	75 (42–90)	<0.001 *
Gender, male	48 (62)	58 (62)	0.997
Diagnosis			
Hilar cholangiocarcinoma	20 (26)	29 (31)	0.018 *
Distal cholangiocarcinoma	39 (51)	28 (30)	
Gallbladder carcinoma	18 (23)	36 (39)	
Serum CA19-9 [U/mL]	91 (1–33,564)	53 (1–2524)	0.133
Hepatectomy	23 (30)	28 (30)	0.973
Clavien-Dindo classification, III–V	35 (45)	31 (33)	0.107
Pathological findings			
Tumor differentiation, well	25 (32)	37 (40)	0.230
Lymphatic invasion	50 (65)	37 (40)	0.001 *
Venous invasion	57 (74)	45 (48)	0.001 *
Perineural invasion	61 (79)	52 (56)	0.005 *
T status, T3 and T4	46 (60)	29 (31)	<0.001 *
N status, N1	48 (62)	23 (25)	<0.001 *
R status, R0	53 (69)	69 (74)	0.255

* Statistical significance (*p* < 0.050). Values in parentheses are the percentages for categorical data or range for continuous data. CA19-9, carbohydrate antigen 19-9.

**Table 2 jcm-10-00925-t002:** Profiles and tumour characteristics of the patients in each group of the matched cohort.

	S-1 Group *n* = 38	Observation Group *n* = 38	*p* Value
Age [y]	72 (52–82)	74 (42–85)	0.640
Gender, male	25 (66)	22 (58)	0.479
Diagnosis			
Hilar cholangiocarcinoma	11 (29)	10 (26)	0.871
Distal cholangiocarcinoma	18 (47)	17 (45)	
Gallbladder carcinoma	9 (24)	11 (29)	
Serum CA19-9 [U/mL]	64 (1–1807)	68 (1–2524)	0.593
Hepatectomy	11 (29)	13 (34)	0.622
Clavien-Dindo classification, III–V	16 (42)	17 (45)	0.817
Pathological findings			
Tumor differentiation, well	14 (37)	14 (37)	1.000
Lymphatic invasion	25 (66)	23 (61)	0.634
Venous invasion	25 (66)	26 (68)	0.807
Perineural invasion	29 (76)	27 (71)	0.602
T status, T3 and T4	20 (53)	21 (55)	0.818
N status, N1	17 (45)	16 (42)	0.817
R status, R0	29 (76)	28 (74)	0.791

Values in parentheses are the percentages for categorical data or range for continuous data. CA19-9, carbohydrate antigen 19-9.

**Table 3 jcm-10-00925-t003:** Univariate and multivariate analysis of overall survival in the matched cohort.

Variable		*n*	Median (Months)	Univariate	Multivariate		
				*p* Value ^†^	HR	95% CI	*p* Value ^‡^
Age [y]	<65	16	86.7	0.274			
	≥65	60	63.8				
Preoperative CA19-9 [U/mL]	<37	23	86.7	0.317			
	≥37	53	58.9				
Clavien-Dindo classification	I–II	43	86.7	0.666			
	III–V	33	63.8				
Differentiation	well	28	86.7	0.134			
	not well	48	58.9				
Lymphatic invasion	no	28	86.7	0.341			
	yes	51	81.5				
Venous invasion	no	25	86.7	0.024 *			
	yes	51	56.7		1.342	0.510–4.102	0.568
Perineural invasion	no	20	86.7	0.007 *			
	yes	56	56.7		6.038	1.709–29.153	0.004 *
T status	T0–T2	35	86.7	0.053			
	T3 and T4	41	58.9				
N status	N0	43	86.7	0.110			
	N1	33	81.5				
R status	R0	57	81.5	0.569			
	R1	19	28.2				
Adjuvant S-1	yes	42	NA	0.003 *			
	no	42	28.2		4.370	1.989–10.298	<0.001 *

* Statistical significance (*p* < 0.050). ^†^ log rank test. ^‡^ Cox proportional-hazards model. NA, not available; HR, hazard ratio; 95% CI, 95% confidence interval.

## Data Availability

The data presented in this study are available in the article.

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
