# Peer review of "The Efficacy of S-1 as Adjuvant Chemotherapy for Resected Biliary Tract Carcinoma: A Propensity Score-Matching Analysis"

_jcm, 2021, doi:10.3390/jcm10050925_

Round 1

Reviewer 1 Report

This is a propensity matched analysis comparing adjuvant S1 versus observation for resected biliary tract malignancy. The main limitations of the paper as described by the authors are the small sample size and the heterogeneity of the primary malignancies included. 

-The OS and RFS analysis of the unmatched group doesn't add to the paper. The adjuvant and observation groups are completely different cohorts. The adjuvant group are more advanced disease. The RFS of the observation group was 81 months vs 25 for S1 group. Although not statically different due to small sample size, but the absolute difference is huge. The difference is obviously driven by the more advanced and aggressive disease selected for adjuvant therapy. I suggest moving this analysis to supplementary or deleting it. 

-The author did subgroup analysis for perineurial invasion and + LN. I suggest they present in two supplementary tables these two subgroups stratified by S1 vs adjuvant so we can understand the baseline characteristics of the cohorts. for example, for +LN patients, show us what are the characteristics for adjuvant vs observation only in this group. While the main cohort is matched, the subgroups may not. 

I suggest that the authors cite other publications on the topic. PMID:29488187 (hilar) and PMID: 32749621 (gallbladder)

Reviewer 2 Report

The authors present a promising study. However, several issues raise concerns.

Introduction

Line 35-36: Authors mention the classification of biliary tract carcinoma according to the Japanese guidelines. It would be reasonable to use a WHO classification since the readers would need to correlate the results to their practice. Thus, ampullary carcinoma does not belong to biliary tract carcinomas, and it can be subclassified indeed in an intestinal-type, pancreaticobiliary type, and a mixed type.

Line 41: Authors say: “ pancreaticoduodenectomy is the most common approach for periampullary carcinoma.” Is the treatment of periampullary carcinomas relevant for the study? It is not biliary tract carcinomas.

Materials and Methods

Line 66: according to the WHO classification, ampullary carcinoma does not belong to the BTC. Which subtype of ampullary carcinoma is included in the study? What type of immunohistochemical markers were used to confirm the subtype of the ampullary carcinoma?

Line 67: R2 resection needs explanation? Is it according to the TNM 8th edition or the Japanese guidelines? What does it mean? Why were these patients excluded?

2.2. Treatment strategy and follow up.

It is not clear why ampullary carcinomas were not included in the study? What criteria were used to classify tumors as ampullary and periampullary? Since correct TNM classification and morphologic/immunohistochemical differentiation of ampullary carcinomas are complicated and require specialized pathologists' evaluation, it would be reasonable to exclude from the study.

2.3. Administration criteria of adjuvant S-1 chemotherapy

The word “microscopic” is unnecessary to use with lymphatic invasion or perineural invasion because it is evident that these two parameters can be evaluated only by microscopy.

Line 109-110: can you specify: “Adjuvant S-1 chemotherapy was performed as long as the patients’ condition were tolerable.”?  What are the criteria for intolerable conditions?

2.4. Statistical analysis

Line 115: Do authors mean UICC TNM classification?

How were specimens evaluated and reported? Was the dataset of the International collaboration on cancer reporting (ICCR) used for reporting the histopathological findings?

Line 131: Here, the authors say that ampullary carcinoma was included. It is not clear periampullary or ampullary tumors were included.

Line 131: lymphatic invasion and perineural invasion are always microscopic because they could be quantified/evaluated only by microscopic evaluation, not during the specimen's grossing.

  1. Results

Lines 140-149: this paragraph is complicated to follow. A schematic presentation of the data could be more understandable.

What does it mean by “staff’s decision”? what was the reason to exclude these patients?

Line 153: criteria to classify the tumor as an ampullary carcinoma? Subtype? Intestinal? Pancreatobiliary subtype? Mixed?

Line 154 and where ever in the text: word “Microscopic” is not necessary

Table 1 and Table 2: not clear why only well-differentiated tumors were included. What about moderate and poorly differentiated tumors? They have a worse prognosis. Were they excluded? What type of hepatectomy? Total or segmental or Hemi-hepatectomy?

3.2. Patient characteristics and survival in the matched cohort

Line 188: What was the total number of patients included in the matched cohort? It seems that there is a miscalculation in the text.

Would you consider excluding ampullary carcinoma from your analysis or present the results on periampullary and ampullary carcinomas in another table/figures? 

Round 2

Reviewer 2 Report

The authors have done a great job revising the document and including distal cholangiocarcinoma in the study. I only have a few additional (minor) comments to make. 

Line 152-153: By explanation, the rationality of the patients' exclusion from the study the mentioning  "the according to staff’s decision" might give an impression that the staff excluded some patients based on the subjective criteria. The authors might consider to remove this expression from the sentence or describe inclusion criteria. 

Line 163: "Hepatectomy was required in 51 patients". What type of surgery was performed in other patients?  Pancreaticoduodenectomy has a high mortality rate as well. Is there any correlation with the outcome? Or do you mean that 51 patients received hepatopancreatoduodenectomy?

Overall comments (lines 160-166): the result section might be difficult to follow because the beginning of the result section describes the whole cohort. The tables are specified according to the groups. 
